# High-Resolution Genotyping of Expressed Equine MHC Reveals a Highly Complex MHC Structure

**DOI:** 10.3390/genes14071422

**Published:** 2023-07-10

**Authors:** Deepali Vasoya, Thomas Tzelos, Lindert Benedictus, Anna Eleonora Karagianni, Scott Pirie, Celia Marr, Charlotta Oddsdóttir, Constanze Fintl, Timothy Connelley

**Affiliations:** 1The Roslin Institute, The Royal (Dick) School of Veterinary Studies, University of Edinburgh, Easter Bush Campus, Roslin EH25 9RG, UK; 2Moredun Research Institute, Pentlands Science Park, Bush Loan, Penicuik EH26 0PZ, UK; 3Faculty of Veterinary Medicine, Utrecht University, Yalelaan 1, 3584 CL Utrecht, The Netherlands; 4Royal (Dick) School of Veterinary Studies, University of Edinburgh, Easter Bush Campus, Roslin EH25 9RG, UK; 5Rossdales Equine Hospital, Cotton End Road, Exning, Newmarket CD8 7NN, UK; 6The Institute for Experimental Pathology at Keldur, University of Iceland Keldnavegur 3, 112 Reykjavík, Iceland; charlotta@hi.is; 7Department of Companion Animal Clinical Sciences, Faculty of Veterinary Medicine and Biosciences, Norwegian University of Life Sciences, P.O. Box 5003, 1432 Ås, Norway

**Keywords:** equine, Major Histocompatibility Complex (MHC), breed diversity, polymorphism, high-throughput sequencing

## Abstract

The Major Histocompatibility Complex (MHC) genes play a key role in a number of biological processes, most notably in immunological responses. The MHCI and MHCII genes incorporate a complex set of highly polymorphic and polygenic series of genes, which, due to the technical limitations of previously available technologies, have only been partially characterized in non-model but economically important species such as the horse. The advent of high-throughput sequencing platforms has provided new opportunities to develop methods to generate high-resolution sequencing data on a large scale and apply them to the analysis of complex gene sets such as the MHC. In this study, we developed and applied a MiSeq-based approach for the combined analysis of the expressed MHCI and MHCII repertoires in cohorts of Thoroughbred, Icelandic, and Norwegian Fjord Horses. The approach enabled us to generate comprehensive MHCI/II data for all of the individuals (*n* = 168) included in the study, identifying 152 and 117 novel MHCI and MHCII sequences, respectively. There was limited overlap in MHCI and MHCII haplotypes between the Thoroughbred and the Icelandic/Norwegian Fjord horses, showcasing the variation in MHC repertoire between genetically divergent breeds, and it can be inferred that there is much more MHC diversity in the global horse population. This study provided novel insights into the structure of the expressed equine MHC repertoire and highlighted unique features of the MHC in horses.

## 1. Introduction

The Major Histocompatibility Complex (MHC) locus contains many genes that are critical to the function of the immune system. This includes the MHC class I (MHCI) and MHC class II (MHCII) genes that encode products that have roles in both the innate and adaptive immune responses—MHCI molecules present peptides to CD8+ “cytotoxic” T cells and also interact with a number of receptors that control Natural Killer (NK) cell activation, while MHCII molecules present peptides to CD4+ “helper” T cells.

Although they perform similar roles with regards to peptide presentation to T cells, the biology of MHCI and MHCII differs in a number of ways. MHCI molecules are composed of a heavy chain with 3 extracellular domains (α1, α2, and α3) non-covalently bound to an invariant β2-microglobulin. The MHCI genes are divided into classical and non-classical genes, which differ in their distribution and function; the former are expressed on nearly all nucleated cells, are highly polymorphic, and predominantly present peptides to “conventional” CD8+ T cells, whereas the latter often exhibit a more restricted distribution, are often mono- or oligo-morphic, and predominantly interact with “non-conventional” T cell subsets, as well as having a greater role in interaction with NK receptors [1]. In MHCI molecules, the peptide binding groove (PBG) is formed by the membrane distal α1 and α2 domains, and in classical MHCI molecules, this is where the majority of the polymorphic variability is located. These polymorphisms alter the structure of the PBG, resulting in variations in the repertoire of peptides that can be bound and presented to T cells by different MHCI alleles. In contrast, MHCII molecules are heterodimers formed by paired α and β chains, both of which contain 2 extra-cellular domains, with the PBG formed by the combined structure of the membrane-distal α1 and β1 domains. For MHCII, most of the polymorphisms in the MHCII genes are also concentrated in the domains that form the PBG. MHCII molecules are divided into 3 main groups—DR, DQ, and DP—and are mainly expressed on specialized antigen-presenting cells such as dendritic cells, macrophages, and B-cells, as well as on activated T cells. Other MHCII molecules, such as DO and DM, serve ancillary roles and do not directly present peptides to T cells. The high level of polymorphism exhibited by classical MHCI and II genes is considered to be a mechanism that ensures that, at a population level, a diverse array of pathogen-derived peptides can be presented to T cells and thus limit the potential for pathogens to exploit variation as a mechanism to evade host immune responses [2]. In humans, a total of 24,703 MHCI alleles and 9719 MHCII alleles have been characterized to date (https://www.ebi.ac.uk/ipd/imgt/hla/about/statistics/, accessed on 15 September 2022).

The evolution of the MHC locus has been complex, with cross-species comparisons illustrating high levels of variation in gene copy number, so that although there is clear evidence of analogous gene families across species, it is not possible to infer the MHCI and MHCII gene content from other species. In the horse (*Equus caballus*), the genomic structure of the MHC locus (on chromosome 20q21) regions containing MHCI and MHCII genes has been described using a combination of bacterial artificial chromosome (BAC) tiling and long read sequencing on the Pacific Biosciences platform [3,4,5]. These studies have identified a high MHCI/MHCII gene content, with 15 MHCI and 15 MHCII loci identified. This includes 4 classical MHCI, 3 non-classical MHCI, and 8 MHCI pseudogene loci; mRNA expression of all classical and non-classical genes was confirmed using loci-specific primers, while a similar approach failed to detect mRNA expression of the pseudogenes [4]. The MHCII loci included 1 DRA locus, 6 DRB loci, and 4 paired DQA and DQB loci; only the DRA locus, 3 DRB loci (DRB1-3), and 3 DQA/DQB paired loci (DQA1-3 and DQB1-3) were considered to encode functional genes, and mRNA expression of each of these has been confirmed using locus-specific primers [5,6]. No MHCII DP genes were identified in the horse (similar to other artiodactyls).

Although evidence of polymorphism in MHC genes has been obtained in numerous studies across a wide range of horse breeds as well as members of the broader Equidae family [4,6,7,8,9,10,11,12,13,14,15,16,17,18,19,20], the characterization of the equine MHC allelic repertoire remains limited. The IPD-MHC database (https://www.ebi.ac.uk/ipd/mhc/group/ELA/, accessed on 15 September 2022) has sequence data for only 48 MHCI, 5 DRA, 20 DRB (combination of DRB1-3 loci), 13 DQA (DQA1-3 loci), and 15 DQB (combination of DQB1-3 loci) alleles.

Prior to the development of “high-throughput sequencing” (HTS) technologies, the generation of MHC sequence data required intensive laboratory work, constraining the development of expansive databases for “non-model” species such as horses. The deployment of HTS technologies has dramatically enhanced the capacity to generate high-resolution data from large cohorts of individuals and has now become standard methodology for human MHC genotyping (as reviewed recently in [21,22,23]). It also affords new opportunities to explore the MHC repertoire of diverse animal species (for example, [24,25,26]). We have recently developed Illumina Miseq-based high-throughput approaches to analyze the combined MHCI and MHCII genotypes of cattle cohorts of various breeds and lineages [27,28,29]. In this study, we developed a parallel system to study the MHC repertoire of horses and applied this to study the MHCI and MHCII alleles expressed in cohorts of Thoroughbred, Icelandic, and Norwegian Fjord Horses. The data generated in this study has characterized the combined MHCI-MHCII haplotypes for a total of 168 animals, identified 152 and 117 novel MHCI and MHCII alleles, and provided novel insights into the profile and composition of expressed equine MHC genes.

## 2. Materials and Methods

### 2.1. Sampling

Samples were collected from Thoroughbreds *(n* = 96), Icelandic (*n* = 35), and Norwegian Fjord (*n* = 37) Horses from the UK, Iceland, and Norway, respectively. All samples were collected with the owner’s informed consent and according to the relevant national legislation, and the study was approved by The Roslin Institute Animal Welfare and Ethics Review Panel. Blood samples were collected by jugular venepuncture into EDTA vacutainers (BD Biosciences, Oxford, UK), and erythrocytes were lysed by incubation in 5x volume of red blood cell (RBC) lysis buffer (0.144 M ammonium chloride/0.175 M Tris pH 7.4) for 5 min at room temperature. The white blood cell (WBC) pellet was washed three times in PBS, total RNA extracted using Tri-reagent (Sigma, Gillingham, UK), and cDNA synthesized using a Reverse Transcription Kit (Promega, Madison, WI, USA), both according to the manufacturer’s instructions.

### 2.2. PCR Amplification and Library Preparation

PCR amplification of the MHCI, DRA, DRB, DQA, and DQB genes was performed using the primers described in Table 1. The design of primers was based on the sequences of MHC alleles available in the IPD database (https://www.ebi.ac.uk/ipd/mhc/group/ELA/, accessed on 6 July 2018). The forward and reverse primers for DRA, DRB, DQA, and DQB were designed to be sited in exon 1 and exon 3, respectively, allowing amplification of the entire polymorphic exon 2. For MHCI, two sets of primers (For1/Rev2.2 and For3.2/Rev1) were designed to be located within exons 2 and 3. The location of these primers permitted the majority of the highly polymorphic exon 2/3 sequence (410 bp) to be amplified. A series of primers incorporating Illumina adaptors and multiplex identifier tags (MID) were obtained from IDT (Leuven, Belgium) to permit the generation of 96 amplicon pools from separate samples for each PCR that could subsequently be multiplexed in a single MiSeq run. PCRs were conducted using the Phusion High-Fidelity PCR Kit (New England Biolabs, Hitchin, UK) with 50 μL reactions composed of Phusion HF amplification buffer, 3% DMSO, 0.2 mM dNTPs, 25 pmol of for and rev primers, 1 U of Phusion Hot Start DNA polymerase, and 1 μL of cDNA. Cycling conditions were 98 °C for 30 s, 30 cycles of 98 °C for 10 s, 61 °C for 30 s, 72 °C for 30 s, and a final extension period of 72 °C for 10 min. Following amplification, 5 μL of PCR product from each sample were pooled, run on a 1% agarose gel, and bands of the appropriate size (~500 bp) were extracted and purified using the Qiagen Gel extraction kit (Qiagen, Manchester, UK). The extracted DNA was subsequently purified using Agencourt AMPure XP beads (Beckham Coulter, High Wycombe, UK) at a *v*/*v* ratio of 1:1 beads to PCR product and quantified using 260/280 nm absorbance readings obtained from a Nanodrop spectrophotometer (Wilmington, DE, USA). The purified amplicons from the MHCI For1/Rev2.2 and For3.2/Rev1, DRA, DRB, DQA, and DQB reactions were mixed at a ratio of 30%:30%:10%:10%:10%:10%, respectively, to generate the library submitted for sequencing. To increase diversity, 10% of PhiX (Illumina, USA) was added to the libraries prior to sequencing.

### 2.3. Sequencing and Bioinformatics Analysis

Libraries were submitted to Edinburgh Genomics, where, after standard quality control procedures, they underwent 300 bp paired-end sequencing on an Illumina MiSeq v3. Sequence reads were segregated based on MID combinations into (up to) 192 datasets, the raw data was assessed for quality (threshold score of >Q_28_), and paired-end sequences were overlapped using FLASH. The data were then processed using a species-specific modified version of a bespoke MHC-genotyping bioinformatics pipeline (available at https://github.com/deepalivasoya/MHCtyping, accessed on 26 April 2023). The downstream analyses of haplotypes and alleles, including calculating frequency, associations, and linkage, were conducted using Perl and Python scripts (https://github.com/deepalivasoya/MHCtyping, accessed on 26 April 2023). Plots were created using ggplot2 in R [30]. Sequence data from previously defined MHC alleles was obtained from the IPD database (https://www.ebi.ac.uk/ipd/mhc/group/ELA/, accessed on 6 July 2018).

### 2.4. Nomenclature of Novel MHC Sequences and Haplotypes

The approach to nomenclature of MHC alleles and haplotypes is in accordance with the IPD-MHC guidelines for nomenclature of non-human MHC sequences generated from NGS data [31]. In brief, nucleotide sequences were translated into amino acids and compared to the data in the IPD-MHC database. If a sequence showed ≤4 amino acid differences from an official designated sequence, then it was considered to be in the same allelic group as that sequence and was given a name to identify it as a novel allele. To avoid potential confusion with official IPD-MHC [31] naming, we used alphabet characters to name novel alleles and synonymous variants (e.g., *Eqca-4*001:AA*). Sequences that showed >4 amino acid differences from any sequence in the IPD-MHC database were considered to represent novel allelic groups and were assigned α-numeric names based on the country of origin of the horses they were first identified in (i.e., gb, is, and no for Great Britain, Iceland, and Norway, respectively) and a number followed by a colon and a double digit number (e.g., *Eqca-MHCI*gb19:01*). MHCI and MHCII haplotypes were defined by recurrent patterns of co-segregation of alleles; to be assigned as a confirmed haplotype, the pattern needed to be repeated in a minimum of 2 animals; if only identified in 1 animal, the putative haplotype was prefixed with “un” (i.e., unconfirmed).

## 3. Results

### 3.1. Design of an Equine MHC Sequencing Platform—Laboratory and Bioinformatics Workflow

The equine MHC typing platform was designed based on the system developed for cattle [27,28,29]. In MHCI, the polymorphic region that forms the peptide binding groove is formed by the α1 and α2 domains encoded by exons 2 and 3, which have a combined length of ~540 bp. As this is beyond the size that can be reliably sequenced on the MiSeq platform, two independent PCRs were developed for the amplification of the MHCI. This approach permitted more of the polymorphic region to be sequenced (410 bp) and also provided some redundancy in case of “allele dropout” (i.e., failure to amplify alleles that have polymorphisms affecting primer binding) as the primers were located within the polymorphic region. In MHCII chains, the polymorphic α1 or β1 domains are encoded by only a single exon (exon 2) of ~270 bp, thus enabling amplicons covering the entire polymorphic region that can be reliably sequenced on the MiSeq platform to be generated from a single PCR (using forward and reverse primers located in the conserved exons 1 and 3, respectively). For DQA, DQB, and DRB, primers were designed to permit amplification of alleles at all loci for these genes. Each primer-pair underwent preliminary validation to ensure the optimized PCR conditions generated a product of expected size from horse PBMC cDNA. The bioinformatics pipeline for the equine MHC analysis (Figure 1) was derived from the pipeline previously established for cattle, with the introduction of a stream to include analysis of Eqca-DRA (BoLA-DRA had not been incorporated into the bovine pipeline as it is considered to be monomorphic).

### 3.2. MHC Diversity in a Population of Thoroughbred Horses

In the absence of access to samples from MHC-defined horses, initial validation work was conducted on a cohort of Thoroughbred horses, as this breed is known to have limited genetic heterogeneity and is the breed in which the MHC has been most intensively studied. Samples were acquired from 96 individuals, including a number of parent-offspring pairs and half-sibling groups.

**Analysis of the MHCI repertoire:** The detailed summary of the data for the Thoroughbred MHCI analysis is provided in Appendix A. A total of 82 MHCI alleles were identified, of which only 13 matched previously described alleles and 69 were novel. Of the novel alleles, 13 represented novel members of previously identified allelic groups, while the remaining 56 represented members of 40 novel allelic groups (Appendix A). Ten different MHCI haplotypes (defined as a set of co-segregating expressed MHCI sequences identified in multiple individuals) were identified in this cohort (HP1.1-HP1.10—Table 2 and Table 3). Sub-types of HP1.1 and HP1.6 were also identified; HP1.1b differed from HP1.1a by the substitution of 5 allelic variants (*Eqca-16*001:AA, Eqca-MHCI*gb6:04, Eqca-7*002:01:AA, Eqca-MHCI*gb13:02* and *Eqca-MHCI*gb11:02* substituting for *Eqca-16*001:01, Eqca-MHCI*gb6:03, Eqca-7*001:01, Eqca-MHCI*gb13:01*, and *Eqca-MHCI*gb11:01:02*), the presence of *Eqca-MHC1*gb38:01* and the absence of *Eqca-4*001:AA* and *Eqca-MHC1*gb9:01*. Similarly, HP1.6b differed from HP1.6a by a combination of both allelic substitutions (*Eqca-MHCI*gb13:01* and *Eqca-7*001:01* substituting for *Eqca-MHCI*gb13:02* and *Eqca-7*002:01:AA*) and the absence of multiple genes (*Eqca-MHCI*gb13:03, Eqca-6*001:01, Eqca-MHCI*gb25:01*, and *Eqca-MHCI*gb31:01*). For the majority of the individuals (81/96, 84.4%) in the cohort, 2 haplotypes were identified, whereas for 13 animals, only a single haplotype was identified (and so these were assumed to be homozygous). One animal (39,234) expressed a single haplotype and a series of additional alleles that were assumed to constitute a second haplotype; however, as this combination of alleles was only observed once, it was assigned as a putative haplotype (unHP1.11—Table 2 and Table 3). The final animal (38,314) expressed a series of alleles that did not conform to any haplotype; it was assumed these alleles formed either a single or two separate haplotypes, but, as these could not be defined, the alleles were left as unassigned (Appendix A). Alleles have been divided into “standard” (i.e., present on a limited number of haplotypes—Table 2) and “universals” (i.e., individual alleles or multiple members of the allele subgroup present in the majority of the haplotypes—Table 3).

There are several notable features of the MHCI repertoires characterized in this cohort of animals. The first of these is the generally large and variable number of sequences identified in each haplotype, ranging from 6 (HP1.10) to 16 (HP1.8 and HP1.4). The second is the wide range of frequency at which the different alleles were observed (Figure 2), which led to low-frequency alleles falling below our defined threshold of resolution (0.2% of reads) in a substantial number of animals. The third is the large proportion of alleles (or closely related allele subtypes) that are expressed on many, and in some cases nearly all, haplotypes and that we consequently termed “universals” (Table 3). This includes *Eqca-7*001:01/002:01:AA* and *Eqca-MHCI*gb11:01:01/11:01:02/11:02* (which match the sequence previously designated as *Eqca-5*001:01*) that have been previously identified as non-classical genes [20], as well as *Eqca-MHCI*g26:01, Eqca-MHCI*g9:01/9:02, Eqca-MHCI*gb13:01/13:02*, and a suite of *Eqca-2*001* alleles. In addition to the “universals”, there are smaller “blocks” of co-segregating sequences shared between a more limited number of haplotypes; for example, *gb1:02, gb4:02*, and *gb27:01* are co-expressed in both HP1.3 and HP1.8 haplotypes. Interestingly, the proportion of reads that the “universals” account for in each individual is high—an average of 50.1% for the For1/Rev2.2 amplicons (ranging from 19.9–88.2%) and 30.9% for the For3.2/Rev1 amplicons (ranging from 6.7–77.0%)—suggesting that they are a numerically substantial component of the expressed MHC repertoires. However, as can be seen in Figure 2, the read frequency for some alleles (e.g., *Eqca-6*001:01*) obtained from the two different MHCI primer pairs significantly diverges, and some alleles are only amplified by one of the primer pairs—consequently, it is not possible to confidently infer mRNA expression level from the read data because of the evident PCR bias. Alleles *Eqca-MHCI*gb9:01, Eqca-1*005:02:01, Eqca-MHCI*gb2:01, Eqca-MHCI*gb4:01*, and *Eqca-MHCI*gb3:02* are described as novel as they are not in the curated IPD database; however, matching sequences have been identified in Ellis et al. (1995) [10], Tallmadge et al. (2010) [20], and Sasaki et al. (2011) [16].

Comparison with previously published data indicated that haplotypes HP1.1a and HP1.6a were broadly consistent with the ELA-A2 and ELA-A3 haplotypes described by Tallmadge et al. (2010) [20] (although in the latter, the Eqca-7*001:01 was replaced with a novel variant *Eqca-7*002:AA*). However, our haplotypes contained substantially higher numbers of expressed genes than had been reported previously (11 vs. 5 for HP1.1a/ELA-A2 and 15 vs. 7 for HP1.6a/ELA-A3). To verify if the novel sequences identified in this study were genuine, those in the HP1.6a/ELA-A3 haplotype were compared to the reference genome, which is derived from an animal known to carry the ELA-A3 haplotype [4]. Fourteen of the 15 sequences were identified in the genome with 100% identity for the full length of the sequenced amplicon (this included *Eqca-7*002:AA*; the exception was *Eqca-MHCI*gb31:01*), confirming that the majority of the sequences obtained in this study represent products of genomic sequences and thus are genuine (Table 4).

**Analysis of the MHCII repertoire:** The detailed summary of the data for the Thoroughbred MHCII analysis is provided in Appendix A. A total of 16, 16, 3, and 19 Eqca-DQA, DQB, DRA, and DRB alleles were identified, of which 6, 6, 0, and 3 (respectively) were novel. One of each of the DQA and DQB sequences represented novel members of previously defined allelic groups, whilst the remaining 5 DQA, 5 DQB, and 3 DRB novel sequences represented members of 4, 5, and 3 novel allelic groups, respectively (Appendix A). Eight different MHCII haplotypes (defined as a set of co-segregating expressed DQA, DQB, DRA, and DRB sequences identified in multiple individuals) were defined from this cohort (HP2.1-HP2.8—Table 5). Sub-types of HP2.2 (HP2.2a–c), HP2.4 (HP2.4a–c), and HP2.6 (HP2.6a–b) were also identified. The HP2.2 subtypes differed in the expressed DRB2 allele, while in HP2.4 and HP2.6 the subtypes differed in the DRA and DRB1 alleles expressed (Table 5). An additional two putative haplotypes were identified in single individuals: unHP2.9 and unHP2.10. With the exception of HP2.5 (for which no DQA2 was identified) and HP2.8 (for which no DRB2 allele was identified), each MHCII haplotype expressed DQA1, DQA2, DQB1, DQB2, DRA, DRB1, and DRB2. Expression of DRB3 could be detected in most haplotypes, but due to the promiscuity of *Eqca-DRB3*001:01:02/Eqca-DRB3*001:01:01*, it was not possible to confirm its presence in HP2.6a, unHP2.9, and unHP2.10 or if it was identified in the individuals expressing these haplotypes due to its presence on the other MHCII haplotypes expressed. No DRB3 allele was identified in any of the HP2.4 subtypes. Thirteen of the animals were homozygous for the MHCII haplotype—10 of these animals were also homozygous for the MHCI haplotype, but the other 3 were MHCI heterozygous (reciprocally, 3 of the MHCI homozygous animals were heterozygous for the MHCII haplotype). Notably, for each of the MHCII genes, there was generally a disparity in the number of reads for each locus; this was greatest for DRB, where the DRB1 locus accounts for 87.7–99.7% of the reads in each haplotype, whereas DRB2 and DRB3 only accounted for 0–12% and 0–3.3% of reads, respectively (Figure 3). For DQB, the DQB1 locus and DQB2 locus accounted for 54.5–99.7% and 0.3–45.5% of reads, respectively, whereas in DQA, there was a slightly more even representation between the DQA1 and DQA2 loci (38.0–100% and 0–62.0%, respectively). Having genotyped both the MHCI and MHCII haplotypes, it was possible to examine the linkage between them. As can be seen in Figure 4A, there is strong linkage between MHCI and MHCII genotypes, with many exhibiting preferential pairings. However, for most MHCI haplotypes identified in the cohort, it appeared that the association with a specific MHCII haplotype was not absolute, and pairings with other MHCII haplotypes were observed. Notably, our data shows an association between HP1.6a and HP2.1, which have previously been shown to be associated with the extended MHCI-MHCII ELA-A3 haplotype [6]. As a corroborative check of our analyses, in members of known half-sibling groups or parent-offspring pairs, the MHCI and MHCII haplotypes identified were consistent with the biological relationships.

### 3.3. Analysis of the MHCI and MHCII Repertoires in Cohorts of Icelandic and Norwegian Fjord Horses

Having been established in a cohort of Thoroughbred horses, we next sought to assess the ability of the MHC genotyping platform to identify the MHC haplotypes in populations of horses in which the MHC repertoire has been less well characterized. For this purpose, we obtained samples from 35 Icelandic and 37 Norwegian Fjord Horses. A detailed summary of the analysis of the MHCI and MHCII repertoires of the animals in these cohorts is provided in Appendix A.

**Analysis of the MHCI repertoire:** From this joint cohort of 72 animals, we identified a total of 132 MHCI genes, of which only 9 matched sequences in the IPD database, another 39 matched sequences identified in the Thoroughbred cohort, and 83 (62.9%) were novel, representing 46 novel allelic groups. As described in the data from the Thoroughbred cohort, although the For1/Rev2.2 and For3.2/Rev1 primers for some alleles gave similar read frequencies (e.g., *Eqca-MHCI*gb15:01*—Figure 5), for other alleles the read frequencies recorded by the two primer pairs were dissimilar or one of the primers suffered “allele dropout” (sometimes even in the same allelic group (e.g., *Eqca-MHCI*gb15:02* and *15:03*—Figure 5)).

We identified 18 MHCI haplotypes (Table 6), 2 of which (HP1.18 and HP1.19) had two subtypes, and an additional 5 putative haplotypes that were only observed in single animals and so were unconfirmed (note that the two HP1.18 subtypes were only observed in single animals and were designated as unconfirmed). Six of the MHCI haplotypes were identified in both the Icelandic and the Norwegian Fjord Horses; only one haplotype (HP1.10) was also identified in the cohort of Thoroughbred horses. However, subtypes of the HP1.7 haplotype were found in an Icelandic horse (unHP1.7b) and in Thoroughbred horses (HP1.7a). Nine animals in the Icelandic/Norwegian Fjord Horses (Ice/Nor) cohort were homozygous for MHCI, sixty-two animals appeared to be heterozygous, and the remaining animal (Norw_037) had a combination of expressed MHCI alleles that could not be designated to any haplotypes, so was left unassigned (similar to animal 38,314 in the Thoroughbred cohort). As with the Thoroughbred cohort, the number of “standard” MHCI genes expressed in each haplotype varied over a wide range (between 2 and 8), but the average number of alleles per haplotype was similar (mean of 6.5 and 5.7 in the Thoroughbred and Ice/Nor cohorts, respectively). The distribution of the “universal” genes amongst the Ice/Nor haplotypes was less consistent than observed in the Thoroughbred cohort; whilst Eqca-MHCI*gb13, *gb26, Eqca-2, and Eqca-7 alleles retained a “universal”-like distribution, being present in all (or nearly all) animals, Eqca-MHCI*gb9 and *gb11 alleles were only observed in 16 and 49 animals, respectively. With the exception of Eqca-2 alleles, it was not possible to reliably assign “universal” alleles to haplotypes, so this was not included in this analysis (Table 6). The reduced prevalence of the Eqca-MHCI*gb9 and *gb11 alleles was associated with a reduction in the proportion of reads assigned to “universal alleles” in the animals in this cohort, with a mean of 37.2% for the For1/Rev2.2 amplicons (range of 9.1–67.6%) and 11.7% for the For3.2/Rev1 amplicons (range of 2.7–24.0%). However, as stated above, clear evidence of PCR bias means that read frequency cannot be used to reliably infer expression levels.

**Analysis of the MHCII repertoire:** A total of 32 DQA, 42 DQB, 4 DRA, and 65 DRB alleles were identified in this cohort. Of these 20, 32, 1, and 49 were novel, respectively, and represented 9 novel DQA, 18 novel DQB, and 23 novel DRB allelic groups (Appendix A). Thirty-three different MHCII haplotypes were identified in this cohort (Table 7). Only 16 of the MHCII haplotypes were identified in multiple individuals and thus could be considered confirmed—this included one haplotype that was also identified in the Thoroughbred cohort (HP2.5) and another that was a variant of the HP2.9 haplotype identified in the Thoroughbred cohort. The remaining 16 haplotypes were novel, of which 5 (HP2.16, HP2.19, HP2.22, HP2.23, and HP2.25) were represented by multiple variants. Some of these variants were only observed in single individuals, so they were assigned as being unconfirmed. An additional 15 haplotypes (HP2.27-HP2.41) were also identified in single individuals and were also considered unconfirmed, although 2 of these haplotypes (unHP2.28 and unHP2.33) were represented by 2 variants (Table 7). The majority of the MHCII haplotypes expressed 2 DQA alleles (18/33), only 4 expressed 3 alleles, while for 4 haplotypes only a single DQA gene was defined. Notably, in 3 haplotypes, no DQA alleles were identified—these were prefixed by a “p” to denote their apparently partial characterization (pHP2.26, unpHP2.40, and unpHP2.41—Table 7). Similarly, the majority of the haplotypes expressed 2 DQB alleles (24/33), whilst only 4 haplotypes were found to express 3 DQB alleles (with the exception of the unHP2.27, these were not the same haplotypes expressing 3 DQA alleles) and 4 haplotypes expressed a single DQB allele (again, the haplotypes expressing single DQA and DQB alleles were not always the same). A single haplotype (unpHP2.40) appeared to express no DQB alleles—this was one of the haplotypes for which no DQA alleles were identified. Most haplotypes express either 2 or 3 DRB alleles, with the presence or absence of the DRB3 allele being the main variable. However, for 5 haplotypes, no DRB2 allele was identified, and, more intriguingly, for 3 haplotypes (HP2.25, unHP2.30, and unHP2.31), no DRB1 allele was identified. Notably, in these latter haplotypes, a suite of additional DRB alleles was identified (Appendix A). These “additional” DRB alleles ranged in number from 2–7 and included different series of co-segregating alleles that were present in subsets of the animals (e.g., *Eqca-DRB*is5:02, Eqca-DRB*is3:01*, and *Eqca-DRB*no4:01* were identified in 4 individuals expressing either unHP2.31 or HP2.25a). This may suggest that within the Icelandic/Norwegian Fjord horse populations, there may be some MHCII haplotypes in which the Eqca-DRB gene structure substantially diverges from that seen in most animals.

As with the Thoroughbred cohort, there is clear evidence that for some of the MHCI haplotypes there is a strong association with specific MHCII haplotypes (e.g., HP1.19a and HP1.19b appear to be associated with HP2.14 and HP2.23a, respectively—Figure 4B). However, most MHCI haplotypes appear to be associated with multiple MHCII haplotypes, and in some cases, MHCI haplotypes do not demonstrate a clear association with any single MHCII haplotype (for example, HP1.24 is associated with HP2.14, HP2.25b, HP2.17, and HP2.11, amongst others, and none are numerically dominant).

## 4. Discussion

The introduction of next-generation sequencing (NGS) approaches has enabled high-resolution and high-throughput MHC genotyping and has fundamentally revolutionized clinical (e.g., [21,22,23]) and research approaches to MHC analysis (e.g., [32,33]). We have recently developed MiSeq-based MHC sequencing approaches to allow the characterization of the MHC genotypes of cattle and applied these to the analysis of MHC diversity in populations of different breeds in various geographical areas [27,28,29]. In this study, we have adapted the NGS MHC genotyping protocol to enable the rapid characterization of the MHCI and MHCII repertoires in horses.

In the first application of the new equine MHC (ELA) typing platform, we analyzed a population of Thoroughbred animals from the UK, which included small parent-offspring or half-sibling subsets. The Thoroughbred is the ideal breed in which to validate the platform for the following reasons: (i) it is the breed in which the ELA has been most intensively studied, e.g., [4,20]; (ii) the genetic diversity in the global Thoroughbred population is restricted due to intensive selective breeding [34,35] with a small effective population size (and as a consequence, increased recurrence of ELA haplotypes—facilitating the identification of co-segregating alleles, which is needed to identify haplotypes); and (iii) there is a genome assembly and other genomic resources available that have been used previously to study the ELA locus on chromosome 20 [3,4,5,36]. Having established the viability of the platform in Thoroughbreds, we next applied the method to a cohort of Icelandic and Norwegian Fjord Horses. These breeds are phylogenetically divergent from the majority of horse breeds, representing a distinct lineage within the global equine population [37,38,39], and so represent a cohort that offers a robust test of the feasibility of applying the system across genetically diverse horse populations.

In total, we identified 152 novel MHCI sequences, representing 86 novel MHCI allelic groups and, for MHCII, a total of 26, 38, 52, and 1 novel DQA, DQB, DRB, and DRA alleles. The abundance of new ELA sequence data generated in this relatively small study provides an indication of the potential for the NGS approach to rapidly expand our knowledge of the equine MHC repertoire. Notably, a high number of novel MHCI alleles were identified in both the Thoroughbred and Icelandic/Norwegian Fjord horse cohorts (69 and 83, respectively), whereas for MHCII, it was only the latter in which a large number of novel alleles were identified (102 vs. 15 novel MHCII alleles identified in the Thoroughbred cohort). Evidence obtained from other studies using microsatellites and other approaches suggests that the MHC repertoires of different breeds show substantial intra- and inter-breed divergence [40,41,42,43,44] and that developing a program to apply this approach to analyzing the MHC repertoires of cohorts of representative animals from across a diverse range of breeds will have rich exploratory value.

In this study, we identified a total of 33 different MHCI haplotypes (of which 5 were assigned as unconfirmed as they were only identified in single animals) and 41 MHCII haplotypes (of which 15 were assigned as unconfirmed). All but 2 of the MHCI and 4 of the MHCII haplotypes appeared to be novel, and by running the MHCI and MHCII analyses in parallel, we were able to also examine the association of MHCI/MHCII in extended MHC haplotypes. A notable feature of both the MHCI and MHCII haplotypes was evidence of haplotype “variants” (i.e., haplotypes where substitution, mutation, and/or insertion/deletion of allele(s) led to the generation of variants of the haplotype); 5/33 of the MHCI haplotypes and 11/41 of the MHCII haplotypes had multiple “variants”. This is consistent with data from previous studies where evidence of recombination events within the MHC [40,41] and allele mutations [45] led to the generation of ELA subtypes. The high frequency of “variants” is a corollary of what appears to be a high degree of instability in the ELA locus; a notable feature of MHCI haplotypes in the Thoroughbred cohort was the recurrent presence of “blocks” of alleles in multiple haplotypes (e.g., *Eqca-6*001:01* and *Eqca-MHCI*gb25:01* in HP1.2, 1.4, 1.6a (but not 1.6b), and 1.8)—suggesting that this section of the ELA locus has been transferred to multiple haplotypes during meiotic recombination. Similarly, the allele content differences between the HP1.6a and HP1.6b haplotypes, which are all associated with a block of genes at the end of the locus (see Table 4), could be explained by a meiotic recombination event. Intriguingly, animal 37,047 was identified as a HP1.1a homozygote but had 3 unassigned alleles: *Eqca-MHCI*gb25:01, Eqca-6*001:01*, and *Eqca-7*002:AA*, which are co-localized in the same region of the ELA locus—perhaps indicating a similar recombination event has generated an additional HP1.1 subtype and the presence of a “recombination hot-spot” within the MHCI region. This data is consistent with previous studies using microsatellite approaches that have identified recombination within the broader MHC locus [40,41,43].

Another notable feature of the MHCI haplotypes was the high but variable number of genes expressed, ranging from 6 to 16 in the Thoroughbred cohort. In previous MHC sequencing work using conventional Sanger sequencing approaches, the number of alleles ascribed to any MHCI haplotype was much more restricted (e.g., in Tallmadge et al. 2010 [20], the average number of MHCI alleles identified per individual was ~5). The ability to map 14/15 of the alleles identified in the HP1.6a haplotype to the ELA locus in the reference genome provided reassurance that the sequences generated in this study were genuine products rather than artifacts. Many of the alleles were represented at very low frequencies in the read data; for example, the average read frequency for *Eqca-MHCI*gb25:01* was 0.11% and 0.18% in the For1 and For3.2 amplicons, respectively. These alleles were initially only identified as being above the read-threshold in a subset of animals (including homozygous animals where there was effectively a “double-dose” gene effect), which subsequently led us to introduce an “allele recovery loop” in the bioinformatics pipeline. These low expression levels probably account for their absence from previous studies using lower-throughput sequencing approaches and suggest that it is feasible that analyses at resolutions even higher than reported here could add additional, very lowly expressed alleles to these haplotypes. Although the evidence of PCR bias (e.g., *Eqca-6*001:01* in Figure 2) precluded us from conducting a formal analysis of inferred expression levels of individual MHCI allele levels, the data generally supports the concept that there is a broad spectrum over which different alleles are expressed (e.g., compare the consistently high read frequency of *Eqca6*001:01* and the low read frequency of *Eqca-4*001:01* in Figure 2). The combination of copy number variation and variation in expression levels of MHCI alleles is reminiscent of the data derived from cattle [27,28,29].

The equine genes that have been labeled as non-classical MHCI have a mixture of features that make this designation uncertain—although they exhibit low levels of polymorphism, they tend to be expressed at levels higher than observed in non-classical MHCI genes of other species [46]. Our data identified a number of MHCI alleles (or subtypes of the same allelic group) that were found in the majority of haplotypes—another feature consistent with a non-classical phenotype. However, as observed previously, these genes appeared to be expressed at high levels (e.g., *Eqca-7** and *Eqca-11** alleles). Due to the functional implications that are inherent in designating a gene as non-classical MHCI [47], we avoided using this term and instead referred to “standard” and “universal” MHCI genes based on their distribution across haplotypes. It should be noted that due to this categorization, *Eqca-2** genes were considered to be “universals”, but *Eqca-6** genes were not—this reverses the non-classical MHCI status previously assigned to these genes [20]. The “universal” genes occupied a high frequency of the reads—in Thoroughbreds, this was ~50% for the For1/Rev2.2 primer pair and ~30% for the For3/Rev1 primer pair amplicons, and some of the individual genes were amongst the most highly represented (e.g., *Eqca-11** and *Eqca-7**), which together suggests the “universals” have functionally important roles. In the Iceland/Norwegian Fjord horse cohort, the “universals” proportion of reads was lower (~40% for For1/Rev2.2 amplicons and ~10% for For3.2/Rev1 amplicons), and it was not possible to reliably assign most “universal” alleles to specific MHCI haplotypes (the exception being the *Eqca-2** alleles), but overall, their contribution to the MHCI repertoire in this cohort remained substantial. Given the unusual characteristics of these “universal” genes, further work to more comprehensively characterize their distribution (i.e., expression profiles on cell subsets) and properties (e.g., immunopeptidomics to identify repertoires of presented peptides [48,49] and studies to define restriction of T cell responses) would be useful to better define their biological function.

Compared to MHCI, there was much more consistency in the number of MHCII loci represented in the data. In most MHCII haplotypes, there was evidence of a single DRA, 2 DQA, 2 DQB, and either 2 or 3 DRB loci. This is generally consistent with the anticipated repertoire of MHCII loci; previous work has identified up to 3 DQA and 3 DQB loci. However, their presence has only been confirmed on some haplotypes, and potential deleterious mutations suggest not all may be expressed [5,6]. In some MHCII haplotypes, the absence of specific DRB/DQA/DQB may be due to allele “drop-out”, and further work may be needed to confirm this. “Allele drop-out” was also seen for MHCI, as evidenced by the detection of specific alleles in only one of the MHCI primer pairs used. Primers were designed based on the limited number of reference sequences available in the IPD-MHC database. The accumulation of more MHCI/II allele sequences (and the chance to use supplementary strategies to interrogate the MHC repertoire of individuals where “allele drop-out” is suspected) could be utilized to modify the primers used in subsequent iterations of the protocol. For a subset of the MHCII haplotypes identified in the Icelandic/Norwegian Fjord horse cohort (HP2.25, unHP2.30, and unHP2.31), there appeared to be a divergent MHCII structure, comprised of the anticipated DRA/DQA/DQB components with an unusual portfolio of DRB genes, which included a single DRB2 allele and then a suite of 4–5 additional DRB genes. The frequency at which these abnormal haplotypes are present in different equid populations, the genomic basis for this genotype, and the possible functional consequences are worth further exploration. As we only performed single PCRs, it is not possible to robustly analyze inferred expression levels of the different DQA/DQB/DRB loci. However, the data indicated that there was a strong hierarchy in DRB, with the DRB1 locus being dominant (~90–100% of DRB reads). Whilst in both DQ genes the DQA1 and DQB1 loci were dominant, this was less pronounced, and, in some cases for DQ, there was near equal read frequency of the DQA1 and DQA2 loci. This suggests that for Eqca-DR molecules, the DRB1 locus is functionally the most important, whereas for Eqca-DQ molecules, the relative importance of the different loci may be more nuanced, with the DQA2/3 and DQB2/3 loci, as well as DQA1 and DQB1, playing a greater role in shaping the expressed MHCII repertoire.

In summary, we have generated and validated a high-resolution and high throughput system for genotyping the MHC in horses. By providing sequencing data for the expressed MHC alleles, this approach has benefits over other approaches (e.g., microsatellites) and can provide novel insights into the repertoire and structure of the MHC repertoire. The MHC influences a number of biological processes that are relevant to the management of equine populations. This includes the immune response (with translational aspects contributing to vaccinology) [15,48,49], disease resistance/susceptibility [13,50,51,52,53,54,55], reproduction [56,57,58,59,60], and novel medical approaches such as mesenchymal stem cell therapy [61,62]. As such, novel tools, such as the one described here, could be utilized in various ways to advance equine healthcare—for example, by enhancing MHC-matching for mating or cellular therapeutics, providing higher resolution capacity to understand linkages of MHC genotypes to specific disease phenotypes, and contributing to a more rationalized approach to the development of new vaccines through enhanced antigen-identification approaches. MHC is also a key marker of genetic diversity that is important for the management of wild/vulnerable equid populations [42,44]. We envision the MHC genotyping method herein described being used to further explore the equine MHC repertoire and facilitate translational work that will have benefits for the health and management of horse populations.

## Figures and Tables

**Figure 1 genes-14-01422-f001:**
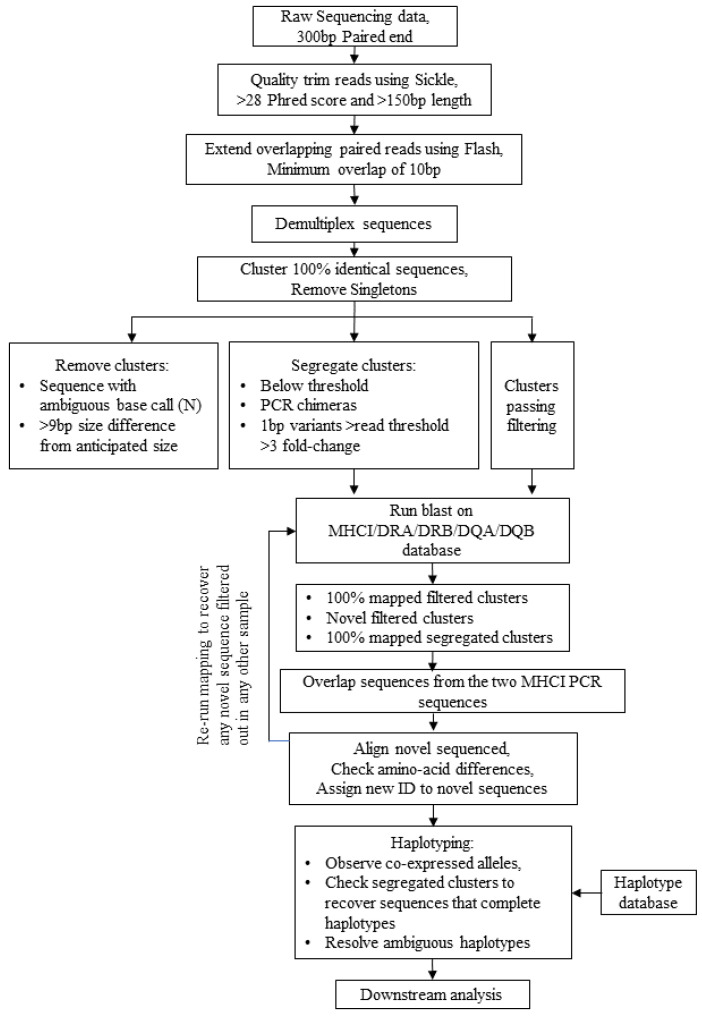
**Schematic workflow of equine MHC genotyping using a MiSeq High-throughput sequencing approach.** An overview of the bespoke bioinformatics pipeline used to analyze the equine MHCI and MHCII data generated using a MiSeq platform, starting with the raw MiSeq data to a final set of MHC alleles and haplotypes.

**Figure 2 genes-14-01422-f002:**
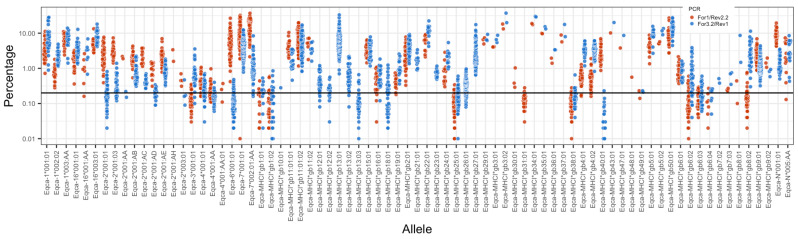
**Read frequency of the MHCI alleles identified in the cohort of Thoroughbred horses.** For each allele (horizontal axis), the percentage of sequencing reads (vertical axis) at which it was identified in the For1/Rev2.2 and For 3.2/Rev1 amplicons in each individual is represented by a blue and red colored dot, as described in the legend. The percentage of sequencing reads is shown on a logarithmic scale. The 0.2% cut-off threshold is shown as a horizontal bar.

**Figure 3 genes-14-01422-f003:**
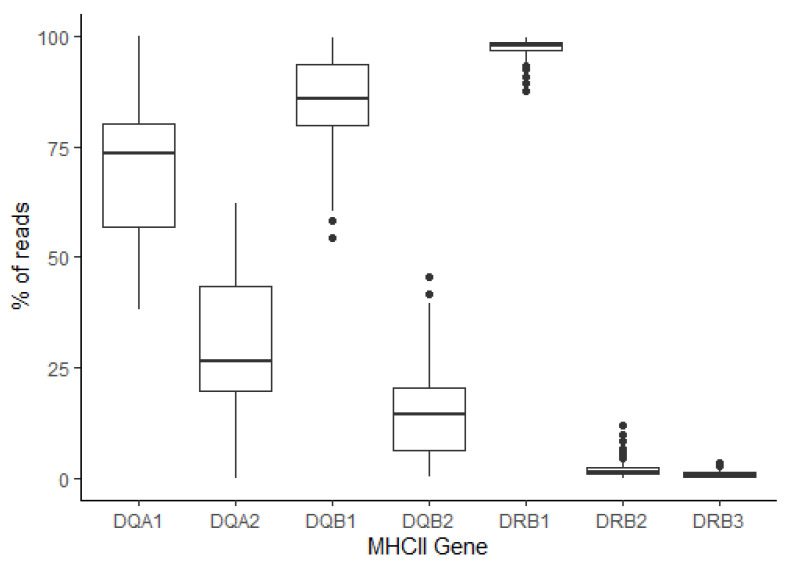
**Read frequency of the (i) DQA, (ii) DQB, and (ii) DRB loci in the Thoroughbred cohort.** The relative frequencies of reads representing alleles for DQA1/DQA2, DQB1/DQB2, and DRB1/DRB2/DRB3 loci are shown. For each gene, the horizontal central line in the box plot represents the median value, the box represents the inter-quartile range, and the vertical lines extending from the box represent 1.5x the inter-quartile range. Individual outliers are represented by dots.

**Figure 4 genes-14-01422-f004:**
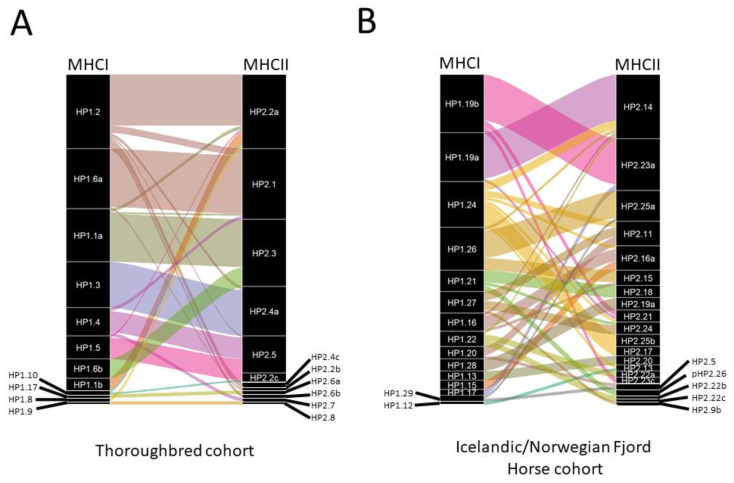
Association between expressed MHCI and MHCII haplotypes in the (**A**) Thoroughbred and (**B**) Icelandic Norwegian Fjord Horse cohorts. In the alluvial plot, the width of the lines linking MHCI and MHCII haplotypes represents the frequency at which these MHCI/MHCII are observed to co-segregate in the cohort.

**Figure 5 genes-14-01422-f005:**
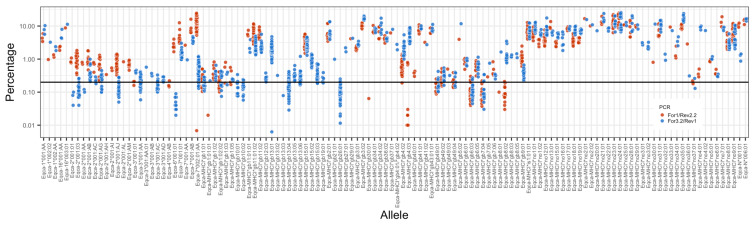
**Read frequency of the MHCI alleles identified in the cohort of Icelandic and Norwegian Fjord Horses.** For each allele (horizontal axis), the percentage of sequencing reads (vertical axis) at which it was identified in the For1/Rev2.2 and For 3.2/Rev1 amplicons in each individual is represented by a blue and red colored dot as described in the legend. The percentage of sequencing reads is shown on a logarithmic scale. The 0.2% cut-off threshold is shown as a horizontal bar.

**Table 1 genes-14-01422-t001:** **Details of ELA-specific primers used in this study.** The size of the amplicon (after the removal of primers) for each primer pair is shown. The combined length of the MHCI sequences generated by the For1/Rev2.2 and For3.2/Rev1 primer pairs is 410 bp. Mismatches between individual alleles and the primer sequences are shown in the final column.

Gene	Orientation	Sequence	Amplicon Size (bp)	Mis-Matches
DRA	For	CAGCTGTCCTGATGAGCTTT	360	
	Rev	AGCCACGTGACATCGATCAC		
DRB	For	GAGGCTCCTGGATGGCAGCT	341	A not G@6 for *Eqca-DRB2*006:01*
	Rev	GTCTTTGCAGGATACACAGT		
DQA	For	GATCCTAAACAGAGCTCTGA	370	
	Rev	AAGACAGATGAGGGTGTTGG		
DQB	For	GGCCTTTGGACAKYAGCT	351	
	Rev	RGATGGGGAGAYGGTCAC		
MHCI	For1	GTYGGCTAYGTGGACGAC	378	T not A@11 and C not T@12 for *Eqca-1*005:02*, *1*008:01, 1*009:01, 18*002:01* and *18*003:01*
	Rev2.2	SCCMTCYAGGTAGKYCCT		A not T@8 for *Eqca-4*001:01* T not C@13 For *Eqca-18*003:01*
MHCI	For3.2	GGGCCGSARTATTGGGA	318	A not G@12 for *Eqca-17*002:02*
	Rev1	CTCCAGGTRTCTGMGGAGC		A not C@5 for *Eqca-6*001:01* C not T@11 for *Eqca-4*001:01* A not T@12 for Eqca-N*001:01

**Table 2 genes-14-01422-t002:** **The “standard” gene content of MHCI haplotypes identified from the cohort of Thoroughbred horses.** For each haplotype, the number of occurrences in the cohort, the total number of expressed “standard genes” and the identity of each allele in the haplotype are shown. For homozygous animals, the haplotype is considered to be expressed twice (i.e., inherited from both the maternal and paternal lineages). The set of alleles assigned to haplotype HP1.11 were only observed in 1 individual animal; consequently, it has been prefixed with “un” to define it as an unconfirmed haplotype. Amongst the “standard alleles”, there are examples where some “blocks” of alleles appear to be present in multiple haplotypes—e.g., *Eqca-MHCI*gb1:02, Eqca-MHCI*gb4:02,* and *Eqca-MHCI*gb27:01* are present in HP1.3 and HP1.8—these are shown in on a grey background. Alleles shown in bold script on a gray background vary between variants of a haplotype (HP1.1a/HP1.1b and HP1.6a/HP1.6b); alleles that differ between these variant haplotypes but appear to not be allelic variants (i.e., are due to differences in allele insertion/deletion) are shown in bold and underlined script. For brevity, the “*Eqca-*” prefix has been removed from all allele names in the Table.

Haplotype.	Number of Occurrences in Cohort	Total Number of Expressed Genes	Standard											
			Number of ‘Standard Genes’ Expressed	Allele	Allele	Allele	Allele	Allele	Allele	Allele	Allele	Allele	Allele	Allele
HP1.1a	30	11	5	** * 4*001:AA * **	*N*001:01*	*MHCI*gb8:02*	** *16*001:01* **	** *MHCI*gb6:03* **						
HP1.1b	7	10	5	** * MHCI*gb38:01 * **	*N*001:01*	*MHCI*gb8:02*	** *16*001:AA* **	** *MHCI*gb6:04* **						
HP1.2	42	12	7	*MHCI*gb12:01*	*MHCI*gb50:01*	*MHCI*gb2:01*					*MHCI*gb4:01*	*MHCI*gb27:01*	*6*001:01*	*MHCI*gb25:01*
HP1.3	27	13	8	*MHCI*gb38:01*	*MHCI*gb40:01*	*1*002:02*	*16*003:01*	*MHCI*gb16:01*		*MHCI*gb1:02*	*MHCI*gb4:02*	*MHCI*gb27:01*		
HP1.4	16	16	10	*MHCI*gb5:01*	*N*005:AA*	*1*003:AA*	*MHCI*gb19:01*	*MHCI*gb6:01*	*MHCI*gb15:01*	*MHCI*gb12:02*	*MHCI*gb4:02*		*6*001:01*	*MHCI*gb25:01*
HP1.5	13	12	7	*MHCI*gb38:01*	*MHCI*gb21:01*	*MHCI*gb22:01*	*1*005:02*	*MHCI*gb23:01*	*MHCI*gb15:01*	*MHCI*gb1:01*				
HP1.6a	34	15	8	** * MHCI*gb31:01 * **	*4*001:01*	*1*001:01*	*3*001:01*	*MHCI*gb6:02*	*MHCI*gb18:01*				** * 6*001:01 * **	** * MHCI*gb25:01 * **
HP1.6b	11	11	5		*4*001:01*	*1*001:01*	*3*001:01*	*MHCI*gb6:02*	*MHCI*gb18:01*					
HP1.7a	2	12	7	*MHCI*gb37:01*	*MHCI*gb36:01*	*MHCI*gb8:01*	*MHCI*gb30:01*			*MHCI*gb1:02*	*MHCI*gb4:02*		*6*001:01*	
HP1.8	3	16	10	*MHCI*gb7:03*	*4*001:AA:01*	*MHCI*gb29:01*	*MHCI*gb3:01*	*MHCI*gb5:02*		*MHCI*gb1:02*	*MHCI*gb4:02*	*MHCI*gb27:01*	*6*001:01*	*MHCI*gb25:01*
HP1.9	2	8	4	*MHCI*gb38:01*	*MHCI*gb7:02*	*MHCI*gb34:01*	*MHCI*gb35:01*							
HP1.10	2	6	2	*MHCI*gb3:02*	*MHCI*gb49:01*									
unHP1.11	1	13	7	*MHCI*gb43:01*	*MHCI*gb47:01*	*MHCI*gb10:01*	*MHCI*gb48:01*	*MHCI*gb30:01*			*MHCI*gb4:02*	*MHCI*gb27:01*		

**Table 3 genes-14-01422-t003:** **The “universal” gene content of MHCI haplotypes identified from the cohort of Thoroughbred horses.** For each haplotype, the number of occurrences in the cohort, the total number of expressed genes, the number of expressed “universal” genes, and the identity of each allele in the haplotype are shown. Alleles shown in bold script on a gray background vary between variants of a haplotype (HP1.1a/HP1.1b and HP1.6a/HP1.6b); alleles that differ between these variant haplotypes but appear to not be allelic variants (i.e., are due to differences in allele insertion/deletion) are shown in bold and underlined script. Due to their promiscuity, it was not possible to confirm that *Eqca-MHCI*gb11:01:02* and *Eqca-MHCI*gb26:01* were expressed in HP1.7, HP1.8, HP1.9, HP1.10 (*Eqca-MHCI*gb26:01* only), and unHP1.11—in these haplotypes, these alleles are shown in white script on a black background. For brevity, the “*Eqca-*” prefix has been removed from all allele names in the Table.

Haplotype.	Number of Occurrences in Cohort	Total Number of Expressed Genes	Universal							
			Number of ‘Universal Alleles’ Expressed	Allele	Allele	Allele	Allele	Allele	Allele	Allele
HP1.1a	30	11	6	** *MHCI*gb11:01:02* **	*2*001:AB*		** *MHCI*gb13:01* **	** *7*001:01* **	** * MHCI*gb9:01 * **	*MHCI*gb26:01*
HP1.1b	7	10	5	** *MHCI*gb11:02* **	*2*001:AB*		** *MHCI*gb13:02* **	** *7*002:01:AA* **		*MHCI*gb26:01*
HP1.2	42	12	5		*2*001:AE*		*MHCI*gb13:01*	*7*001:01*	*MHCI*gb9:01*	*MHCI*gb26:01*
HP1.3	27	13	5	*MHCI*gb11:01:02*	*2*001:03*		*MHCI*gb13:01*	*7*001:01*		*MHCI*gb26:01*
HP1.4	16	16	6	*MHCI*gb11:01:01*	*2*001:AD*		*MHCI*gb13:01*	*7*001:01*	*MHCI*gb9:01*	*MHCI*gb26:01*
HP1.5	13	12	5	*MHCI*gb11:01:02*	*2*001:AC*		*MHCI*gb13:01*	*7*002:01:AA*		*MHCI*gb26:01*
HP1.6a	34	15	7	*MHCI*gb11:01:02*	*2*001:01*	** * MHCI*gb13:03 * **	** *MHCI*gb13:02* **	** *7*002:01:AA* **	*MHCI*gb9:01*	*MHCI*gb26:01*
HP1.6b	11	11	6	*MHCI*gb11:01:02*	*2*001:01*		** *MHCI*gb13:01* **	** *7*001:01* **	*MHCI*gb9:01*	*MHCI*gb26:01*
HP1.7a	2	12	5	*MHCI*gb11:01:02*	*2*001:AH*			*7*001:01*	*MHCI*gb9:02*	*MHCI*gb26:01*
HP1.8	3	16	6	*MHCI*gb11:01:02*	*2*003:01*		*MHCI*gb13:01*	*7*001:01*	*MHCI*gb9:01*	*MHCI*gb26:01*
HP1.9	2	8	4	*MHCI*gb11:01:02*	*2*001:AA*			*7*002:01:AA*		*MHCI*gb26:01*
HP1.10	2	6	4	*MHCI*gb11:01:01*			*MHCI*gb13:01*		*MHCI*gb9:01*	*MHCI*gb26:01*
unHP1.11	1	13	6	*MHCI*gb11:01:02*	*2*003:01*		*MHCI*gb13:01*	*7*001:01*	*MHCI*gb9:02*	*MHCI*gb26:01*

**Table 4 genes-14-01422-t004:** Comparison of the MHCI sequences identified in the HP1.6a haplotype with those present in the genome of an individual carrying the ELA-A3 haplotype. For each sequence identified in the HP1.6a haplotype (column 1), the matching cluster identified in Tallmadge et al. (2005) [4] and the functionality of those clusters inferred in that study are shown (columns 2 and 3). For each allele, the location of the matching sequence in the genome (columns 4–6), the orientation of the genes (F—forward, R—R; column 7), matching coordinates of the allele (query) sequence (columns 8–9), the length of the alignment, the score, E-value, and percentage identity are shown (columns 10–13). As the allele sequences covered parts of exons 2 and 3, for each allele, there are two matching genomic sequences (separated by an intron). With the exception of *Eqca-MHCI*gb31:01*, all allele sequences from this study showed 100% identity with a sequence in the genome over the full length of the allele sequence (410 bp for most alleles; for *Eqca-MHCI*gb18:01, Eqca-MHCI*gb13:03, Eqca-MHCI*gb13:02*, and *Eqca-MHCI*gb26:01*, the length of the allelic sequence was only 318 bp as only the For3.2/Rev1 primer pair resulted in the generation of an amplicon). Automated prediction using the Gnomon algorithm has identified genome sequences matching *Eqca-MHCI*gb18:01, Eqca-MHCI*gb13:02,* and *Eqca-MHCI*gb13:03* in the genome (column 14). Note (i) that a sequence matching *Eqca-MHCI*gb9:01* was identified in Ellis et al. (1995) [10]; (ii) in Tallmadge et al. (2005) [4] *Eqca-MHCI*gb25:01* and *gb26:01* were identified as pseudogenes, and no mRNA was identified—in this study, both were identified, although at low levels and, in the case of **gb26:01*, by only one of the primer pairs. For brevity, the “*Eqca-*” prefix has been removed from all allele names in the Table.

Allele	Cluster	Inferred Function	Chr.	Genomic Location—Start	Genomic Location—Stop	Orientation	Query Start	Query End	Length	Score	E-Val	% ID	Automated Gnomon Prediction
*MHCI*gb11:01:02*	3-5	Expressed non-classical	20	29901699	29901880	F	1	182	182	360	4 × 10^−97^	100	
			20	29902124	29902353	F	181	410	230	455	1 × 10^−125^	100	
*MHCI*gb18:01*			20	29958884	29958973	F	1	90	90	178	1 × 10^−42^	100	Yes
			20	29959206	29959437	F	87	318	232	459	5 × 10^−127^	100	
*MHCI*gb6:02*			20	30001109	30001290	F	1	182	182	360	4 × 10^−97^	100	
			20	30001526	30001755	F	181	410	230	455	1 × 10^−125^	100	
*4*001:01*	3-4	Expressed classical	20	30070552	30070733	F	1	182	182	360	4 × 10^−97^	100	
			20	30070970	30071199	F	181	410	230	455	1 × 10^−125^	100	
*3*001:01*	3-3	Expressed classical	20	30225873	30226054	F	1	182	182	360	4 × 10^−97^	100	
			20	30226290	30226519	F	181	410	230	455	1 × 10^−125^	100	
*2*001:01*	3-2	Expressed classical	20	30277079	30277260	F	1	182	182	360	4 × 10^−97^	100	
			20	30277488	30277717	F	181	410	230	455	1 × 10^−125^	100	
*1*001:01*	3-1	Expressed classical	20	30335949	30336130	F	1	182	182	360	4 × 10^−97^	100	
			20	30336359	30336588	F	181	410	230	455	1 × 10^−125^	100	
*MHCI*gb13:03*			20	31136277	31136506	R	89	318	230	455	7 × 10^−126^	100	Yes
			20	31143897	31143986	R	1	90	90	178	1 × 10^−42^	100	
*MHCI*gb13:02*			20	31256964	31257193	R	89	318	230	455	7 × 10^−126^	100	Yes
			20	31264867	31264956	R	1	90	90	178	1 × 10^−42^	100	
*7*002:AA*	3-7	Expressed non-classical	20	31524136	31524365	R	181	410	230	455	1 × 10^−125^	100	
			20	31524606	31524787	R	1	182	182	360	4 × 10^−97^	100	
*MHCI*gb9:01*			20	31884461	31884642	F	1	182	182	360	4 × 10^−97^	100	
			20	31884876	31885105	F	181	410	230	455	1 × 10^−125^	100	
*MHCI*gb26:01*	3-11	Pseudogene	20	31981174	31981403	R	89	318	230	455	7 × 10^−126^	100	
			20	31981615	31981704	R	1	90	90	178	1 × 10^−42^	100	
*6*001:01*	3-6	Expressed non-classical	20	32015995	32016224	R	181	410	230	455	1 × 10^−125^	100	
			20	32016463	32016644	R	1	182	182	360	4 × 10^−97^	100	
*MHCI*gb25:01*	3-10	Pseudogene	20	32060222	32060451	R	181	410	230	455	1 × 10^−125^	100	
			20	32060689	32060870	R	1	182	182	360	4 × 10^−97^	100	
*MHCI*gb31:01*			20	30336359	30336550	F	181	372	192	316	4 × 10^−84^	95.83	
			20	31823220	31823401	F	1	182	182	344	2 × 10^−92^	98.90	

**Table 5 genes-14-01422-t005:** The gene content of MHCII haplotypes identified from the cohort of Thoroughbred horses. For each haplotype, the number of occurrences in the cohort, the number of expressed genes, and the identity of each allele in the haplotype are shown. For homozygous animals, the haplotype is considered to be expressed twice (i.e., inherited from both the maternal and paternal lineages). MHCII haplotypes reported in Miller et al. (2017) [6] that matched the MHCII genotypes described herein are shown (alleles that vary between the previously reported haplotype and the data presented herein are shown in bold script). The set of alleles assigned to haplotypes HP2.4b, HP2.9, and HP2.10 were only observed in 1 individual animal; consequently, they have been prefixed with “un” to define them as unconfirmed haplotypes. Alleles shown in bold script on a gray background vary between variants of the haplotype (HP2.2, HP2.4, and HP2.6). Due to its promiscuity, it was not possible to confirm that *Eqca-DRB3*001:01:02/Eqca-DRB3*001:01:01* was expressed in HP2.6a, unHP2.9, and unHP2.10—in these haplotypes, these alleles are shown in white script on a black background. In HP2.5, no DQA2 allele was identified, while in HP2.8, no DRB2 allele was identified, and in HP2.4a-c, no DRB3 allele was identified. For brevity, the “*Eqca-*” prefix has been removed from all allele names in the Table.

Haplotype	Number of Occurences in Cohort	Number of Expressed Genes	Previousy Defined	DQA1	DQA2	DQB1	DQB2	DRA	DRB1	DRB2	DRB3
HP2.1	40	8	A3	*DQA1*001:01*	*DQA2*001:01*	** *DQB*gb2:01* **	** *DQB*gb3:01* **	*DRA*001:01/* *DRA*001:04*	*DRB1*001:01*	*DRB2*002:01*	*DRB3*001:01:02* */DRB3*001:01:01*
HP2.2a	42	8		*DQA1*002:03*	*DQA*gb1:01*	*DQB1*006:01*	*DQB2*003:01*	*DRA*001:02*	*DRB1*003:01*	** *DRB2*001:01/* ** ** *DRB2*001:02* **	** *DRB3*001:01:02/* ** ** *DRB3*001:01:01* **
HP2.2b	2	8		*DQA1*002:03*	*DQA*gb1:01*	*DQB1*006:01*	*DQB2*003:01*	*DRA*001:02*	*DRB1*003:01*	** *DRB2*002:01* **	** *DRB3*001:01:02/* ** ** *DRB3*001:01:01* **
HP2.2c	5	8		*DQA1*002:03*	*DQA*gb1:01*	*DQB1*006:01*	*DQB2*003:01*	*DRA*001:02*	*DRB1*003:01*	** *DRB2*005:01* **	** *DRB3*002:01/* ** ** *DRB3*002:02* **
HP2.3	38	8	A2	*DQA1*002:01*	** *DQA*gb4:01* **	*DQB1*002:01*	*DQB2*002:01*	*DRA*001:01/* *DRA*001:04*	*DRB1*002:01*	*DRB2*003:01*	*DRB3*001:01:02/* *DRB3*001:01:01*
HP2.4a	28	7		*DQA1*003:01*	*DQA2*002:01*	*DQB1*004:01*	*DQB2*004:01*	** *DRA*001:03* **	** *DRB1*004:01* **	*DRB2*004:01*	
unHP2.4b	1	7	A9	*DQA1*003:01*	*DQA2*002:01*	*DQB1*004:01*	*DQB2*004:01*	** *DRA*001:01* ** ** */DRA*001:04* **	** *DRB1*004:01* **	*DRB2*004:01*	
HP2.4c	3	7		*DQA1*003:01*	*DQA2*002:01*	*DQB1*004:01*	*DQB2*004:01*	** *DRA*001:02* **	** *DRB1*003:01* **	*DRB2*004:01*	
HP2.5	21	7	A10	*DQA1*004:01*		*DQB1*005:01*	*DQB2*005:01*	*DRA*001:01/* *DRA*001:04*	*DRB1*005:01*	*DRB2*001:01/* *DRB2*001:02*	*DRB3*002:01* */DRB3*002:02*
HP2.6a	3	7		*DQA1*002:02*	*DQA2*002:01*	*DQB1*007:01*	*DQB2*004:01*	** *DRA*001:01/* ** ** *DRA*001:04* **	** *DRB*gb1:01* **	*DRB*gb2:01*	*DRB3*001:01:02/* *DRB3*001:01:01**
HP2.6b	2	8		*DQA1*002:02*	*DQA2*002:01*	*DQB1*007:01*	*DQB2*004:01*	** *DRA*001:02* **	** *DRB1*003:01* **	*DRB*gb2:01*	*DRB3*001:01:02/* *DRB3*001:01:01*
HP2.7	3	8		*DQA1*002:05*	*DQA*gb3:01*	*DQB1*003:AA*	*DQB2*003:01*	*DRA*001:01/* *DRA*001:04*	*DRB1*007:01/* *DRB1*007:02*	*DRB2*002:01*	*DRB3*001:01:02/* *DRB3*001:01:01*
HP2.8	2	7		*DQA*gb2:01*	*DQA2*002:AA*	*DQB*gb1:01*	*DQB*gb4:01*	*DRA*001:03*	*DRB*gb2:01*		*DRB3*002:01* */DRB3*002:02*
unHP2.9a	1	7		*DQA1*005:01*	*DQA*gb1:04*	*DQB*gb5:01*	*DQB2*003:01*	*DRA*001:03*	*DRB1*008:01*	*DRB2*006:01*	*DRB3*001:01:02/* *DRB3*001:01:01**
unHP2.10	1	7		*DQA1*002:02*	*DQA*gb1:01*	*DQB1*003:01*	*DQB2*003:01*	*DRA*001:03*	*DRB*gb3:01*	*DRB2*001:01/* *DRB2*001:02*	*DRB3*001:01:02/* *DRB3*001:01:01**

**Table 6 genes-14-01422-t006:** **The gene content of MHCI haplotypes identified from the Icelandic and Norwegian Fjord horses.** For each haplotype, the number of occurrences in Icelandic and Norwegian Fjord horses and the combined cohort, the number of expressed “standard” alleles, and the identity of each allele in the haplotype are shown. For homozygous animals, the haplotype is considered to be expressed twice (i.e., inherited from both the maternal and paternal lineages). Haplotypes prefixed with “un” were only identified in individual animals and so are considered as unconfirmed haplotypes. No “blocks” of alleles co-segregating in multiple haplotypes as observed in the Thoroughbred cohort, were observed in this cohort. Alleles shown in bold script on a grey background vary between variants of the haplotype (HP1.118a/HP1.18b and HP1.19a/HP1.19b and unHP1.7b/HP1.7a (see Table 2)); alleles that differ between these variant haplotypes but appear to not be allelic variants (i.e., are due to differences in allele insertion/deletion) are shown in bold and underlined script. Due to their promiscuity, it was not possible to reliably identify the association of most “universal” alleles with specific haplotypes (see text)—the exception was the Eqca-2*001 alleles, which are shown in the table. For brevity, the “*Eqca-*” prefix has been removed from all allele names in the Table.

Haplotype	Number of Occurences in Icelandic Ponies	Number of Occurences in Norwegian Fjord Horses	Standard									Universal
			Number of ‘Standard Alleles’ Expressed	Allele	Allele	Allele	Allele	Allele	Allele	Allele	Allele	Allele
HP1.10	6	4	2	*MHCI*gb3:02*	*MHCI*gb49:01*							
HP1.12	2	1	7	*1*001:AA*	*MHCI*no34:01*	*MHCI*gb1:02*	*MHCI*no6:01*	*MHCI*gb6:05*	*4*001:AB*	*3*001:01*		*2*001:01*
HP1.13	3	0	6	*N*006:01*	*MHCI*no5:01*	*MHCI*no3:02*	*MHCI*gb8:03*	*MHCI*gb6:05*	*6*001:01*			*2*001:AC*
HP1.14	2	0	7	*MHCI*no25:01*	*MHCI*gb34:01*	*MHCI*no3:02*	*MHCI*gb40:01*	*MHCI*gb6:05*	*1*002:AA*	*MHCI*gb1:02*		*2*001:AA*
HP1.15	3	0	5	*MHCI*no17:01*	*MHCI*no1:02*	*MHCI*gb1:02:02*	*MHCI*gb49:02*			*3*001:AA*		
HP1.16	8	0	6	*MHCI*no16:01*	*MHCI*no28:01*	*MHCI*gb15:01*	*MHCI*gb34:02*	*MHCI*gb6:03*	*MHCI*gb49:04*			
HP1.17	0	2	5	*MHCI*no29:01*	*MHCI*no1:02*	*MHCI*no3:01*	*MHCI*gb8:05*	*MHCI*gb6:05*				
unHP1.18a	1	0	4	*MHCI*gb1:01*	** *MHCI*no2:02* **	*MHCI*gb34:01*	*MHCI*no32:01*					*2*001:AI*
unHP1.18b	1	0	4	*MHCI*gb1:01*	** *MHCI*no2:01* **	*MHCI*gb34:01*	*MHCI*no32:01*					*2*001:AI*
HP1.19a	0	16	6	*MHCI*no24:01*	*MHCI*no9:01*	** * MHCI*gb2:01 * **	*MHCI*gb4:03*		*MHCI*gb15:03*	** * MHCI*gb10:02 * **		*2*001:AJ*
HP1.19b	1	19	7	*MHCI*no24:01*	*MHCI*no9:01*	** * MHCI*no12:01 * **	*MHCI*gb4:03*	** * MHCI*gb6:03 * **	*MHCI*gb15:03*	** * 3*001:AA * **		*2*001:AJ*
HP1.20	3	3	5	*MHCI*gb41:01*	*MHCI*no31:01*	*MHCI*gb4:02*	*MHCI*gb1:01*		*MHCI*gb12:03*			*2*003:01*
HP1.21	0	9	4	*MHCI*no35:01*	*MHCI*no19:01*	*MHCI*gb1:01*	*MHCI*gb10:01*					*2*001:AM*
HP1.22	0	5	4	*MHCI*gb43:01:01*	*MHCI*gb36:02*	*MHCI*gb3:01*	*MHCI*gb4:02*					*2*003:01*
HP1.23	3	0	4	*MHCI*no31:01*	*MHCI*no7:01*	*MHCI*gb4:02*	*MHCI*gb12:03*					*2*003:01*
HP1.24	19	0	7	*MHCI*no23:01*	*MHCI*is1:01:01*	*MHCI*gb4:02*	*MHCI*gb15:01*	*MHCI*no8:01*	*MHCI*gb1:02:02*	*3*001:AC*		*2*001:03*
HP1.25	3	0	6	*MHCI*no13:01*	*MHCI*no4:01*	*MHCI*no22:01*	*MHCI*gb7:06*	*MHCI*gb1:05*	*MHCI*gb1:03*			*2*001:03*
HP1.26	5	10	7	*MHCI*no1:01*	*MHCI*no18:01*	*MHCI*gb8:02*	*MHCI*gb15:02*	*MHCI*gb49:02*	*3*001:AA*	*MHCI*gb7:04*		*2*001:AB*
HP1.27	1	2	5	*N*001:01*	*MHCI*gb8:02*	*16*001:AA*	*2*001:AB*	*MHCI*gb27:01*				
HP1.28	5	0	8	*MHCI*no21:01*	*MHCI*no15:01*	*MHCI*gb6:01*	*MHCI*gb19:01*	*MHCI*gb15:02*	*3*001:AD*	*MHCI*gb1:02*	*6*001:01*	*2*001:03*
unHP1.30	1	0	7	*16*003:01*	*1*002:02*	*3*001:AA:01*	*MHCI*gb15:03*	*MHCI*gb40:01*	*MHCI*gb12:03*	*MHCI*gb1:01:01*		*2*001:03*
unHP1.31	1	0	8	*MHCI*gb5:02*	*MHCI*gb29:01*	*MHCI*gb1:02*	*MHCI*gb15:01*	*MHCI*gb4:02*	*MHCI*gb7:05*	*MHCI*gb1:05*	*3*001:AB*	
unHP1.32	0	1	6	*MHCI*no4:02*	*MHCI*no20:01*	*MHCI*gb41:02*	*MHCI*gb21:01*	*MHCI*gb1:01*	*6*001:01*			*2*001:AC*
unHP1.33	1	0	7	*MHCI*no30:01*	*MHCI*no33:01*	*MHCI*gb15:01*	*MHCI*no36:01*	*MHCI*gb49:03*	*MHCI*gb7:06*	*MHCI*no37:01*		*2*001:AL*
unHP1.7b	1	0	7	*MHCI*gb37:01*	** *MHCI*gb36:02* **	*MHCI*gb8:01*	*MHCI*gb30:01*	*MHCI*gb1:02*	*MHCI*gb4:02*	** * MHCI*gb10:01 * **		*2*001:AH*

**Table 7 genes-14-01422-t007:** **The gene content of MHCII haplotypes identified from Icelandic and Norwegian Fjord horses.** For each haplotype, the number of occurrences in Icelandic and Norwegian Fjord Horses, the number of expressed genes, and the identity of each allele in the haplotype are shown. For homozygous animals, the haplotype is considered to be expressed twice (i.e., inherited from both the maternal and paternal lineages). Haplotypes prefixed with “un” were only identified in individual animals and so are considered unconfirmed haplotypes. Alleles shown in bold script on a grey background vary between variants of the haplotype (HP2.9a/HP2.9b (see Table 5), HP2.16a/unHP2.16b, HP2.19a/HP2.19b, HP2.22a/HP2.22b/HP2.22c, HP2.23a/unHP2.23b/HP2.23c/HP2.23d, HP2.25a/HP2.25b/unHP2.25c, unHP2.28a/unHP2.28b, and unHP2.33a/unHP2.33b). It was assumed that the DRA allele expressed in unHP2.28b was *Eqca-DRA*001:01/Eqca-DRA*001:04,* as this was the only DRA allele identified in the individual in which this haplotype was observed (shown as white script on a black background). For brevity, the “*Eqca-*” prefix has been removed from all allele names in the Table.

Haplotype	Number of Occurences Combined in Icelandic Horses	Number of Occurences in Norwegian Fjord Horses	Number of Expressed Genes	DQA1	DQA2	DQA3	DQB1	DQB2	DQB3	DRA	DRB1	DRB2	DRB3
HP2.5	0	2	7	*DQA1*004:01*			*DQB1*005:01*	*DQB2*005:01*		*DRA*001:01/* *DRA*001:04*	*DRB1*005:01*	*DRB2*001:01/* *DRB2*001:02*	*DRB3*002:01/* *DRB3*002:02*
HP2.9b	3	0	8	*DQA1*005:01*	*DQA*gb1:04*		*DQB*gb5:01*	*DQB2*003:01*		** *DRA*001:01/* ** ** *DRA*001:04* **	** *DRB1*007:01/* ** ** *DRB1*007:02* **	*DRB2*006:01*	** *DRB3*002:01/* ** ** *DRB3*002:02* **
HP2.11	9	2	6	*DQA*no1:01*	*DQA1*002:01:AA*		*DQB1*007:AA*	*DQB*no1:01*		*DRA*001:01/* *DRA*001:04*	*DRB*is10:01*		
HP2.12	3	0	8	*DQA1*002:03*	*DQA*gb1:01*		*DQB*gb1:03*	*DQB2*003:AA*		*DRA*001:01/* *DRA*001:04*	*DRB*is6:01*	*DRB2*002:01:AA*	*DRB3*001:01:02/* *DRB3*001:01:01*
HP2.13	2	1	7	*DQA*gb1:06*			*DQB*is3:01*	*DQB2*002:AA*		*DRA*001:03*	*DRB*gb3:01*	*DRB*gb2:01*	*DRB3*002:01/* *DRB3*002:02*
HP2.14	4	17	6	*DQA1*002:02*	*DQA*gb1:02*		*DQB*no9:01*	*DQB2*003:01*		*DRA*001:01/* *DRA*001:04*	*DRB*gb1:01*		
HP2.15	6	0	7	*DQA*gb2:01*	*DQA2*002:AA*		*DQB*gb1:01*	*DQB*gb4:01*		*DRA*001:01/* *DRA*001:04*	*DRB*is6:01*		*DRB3*001:01:02/* *DRB3*001:01:01*
HP2.16a	8	0	8	*DQA1*002:AA*	*DQA*gb1:05:04*		*DQB1*006:AA*	*DQB2*002:AB*		*DRA*001:01/* *DRA*001:04*	*DRB*is8:01*	** *DRB2*002:01:AA* **	*DRB3*002:01/* *DRB3*002:02*
unHP2.16b	1	0	7	*DQA1*002:AA*	*DQA*gb1:05:04*		*DQB1*006:AA*	*DQB2*002:AB*		*DRA*001:01/* *DRA*001:04*	*DRB*is8:01*	** *DRB2*002:AA* **	
HP2.17	3	0	6	*DQA*is4:01*	*DQA*gb3:01*		*DQB*is4:01*			*DRA*001:01/* *DRA*001:04*	*DRB*is9:01*	*DRB2*004:01*	
HP2.18	0	4	7	*DQA*no1:01*			*DQB*no7:01*	*DQB*no4:01*		*DRA*001:01/* *DRA*001:04*	*DRB*no7:01*	*DRB2*003:01*	*DRB3*002:01/* *DRB3*002:02*
HP2.19a	4	0	8	*DQA*is5:01*	*DQA*gb1:01*	*DQA*is3:01*	*DQB*is8:01*	*DQB*is2:01*		*DRA*001:01/* *DRA*001:04*	** *DRB1*002:01* **	** *DRB2*005:01* **	
HP2.19b	2	0	8	*DQA*is5:01*	*DQA*gb1:01*	*DQA*is3:01*	*DQB*is8:01*	*DQB*is2:01*		*DRA*001:01/* *DRA*001:04*	** *DRB1*001:01* **	** *DRB2*002:01* **	
HP2.20	4	0	8	*DQA1*002:01*	*DQA*gb1:02*		*DQB1*007:AB*	*DQB2*003:01*		*DRA*001:01/* *DRA*001:04*	*DRB*no7:01*	*DRB2*001:01/* *DRB2*001:02*	*DRB3*002:01/* *DRB3*002:02*
HP2.21	1	3	7	*DQA*no3:01*	*DQA2*002:01*		*DQB*no5:01*	*DQB2*004:01*		*DRA*001:01/* *DRA*001:04*	*DRB*no8:01*	*DRB2*002:01*	
HP2.22a	0	2	9	*DQA*no2:01*	*DQA*gb3:01*	*DQA*no5:01*	*DQB*no8:01*	*DQB*no10:01*		*DRA*001:01/* *DRA*001:04*	** *DRB*no1:01:01* **	*DRB*no6:01*	*DRB3*002:01/* *DRB3*002:02*
unHP2.22b	0	1	9	*DQA*no2:01*	*DQA*gb3:01*	*DQA*no5:01*	*DQB*no8:01*	*DQB*no10:01*		*DRA*001:01/* *DRA*001:04*	** *DRB*no1:03* **	*DRB*no6:01*	*DRB3*002:01/* *DRB3*002:02*
unHP2.22c	0	1	9	*DQA*no2:01*	*DQA*gb3:01*	*DQA*no5:01*	*DQB*no8:01*	*DQB*no10:01*		*DRA*001:01/* *DRA*001:04*	** *DRB*no1:02* **	*DRB*no6:01*	*DRB3*002:01/* *DRB3*002:02*
HP2.23a	0	17	6	*DQA*no1:01*			** *DQB*no1:01* **	*DQB*no6:01*		*DRA*001:01/* *DRA*001:04*	*DRB1*002:01*		** *DRB3*002:01/* ** ** *DRB3*002:02* **
unHP2.23b	0	1	7	*DQA*no1:01*			** *DQB*no1:01* **	*DQB*no6:01*		*DRA*001:01/* *DRA*001:04*	*DRB1*002:01*	** *DRB2*002:01* **	** *DRB3*001:01:02/* ** ** *DRB3*001:01:01* **
HP2.23c	0	2	7	*DQA*no1:01*			** *DQB*no1:01* **	*DQB*no6:01*		*DRA*001:01/* *DRA*001:04*	*DRB1*002:01*	** *DRB2*002:01* **	** *DRB3*002:AA* **
unHP2.23d	1	0	7	*DQA*no1:01*			** *DQB*no1:03* **	*DQB*no6:01*		*DRA*001:01/* *DRA*001:04*	*DRB1*002:01*	** *DRB2*002:01* **	** *DRB3*001:01:02/* ** ** *DRB3*001:01:01* **
HP2.24	0	5	8	*DQA1*002:05*	*DQA2*002:AA*		*DQB1*003:01*	*DQB*no1:02*		*DRA*001:03*	*DRB1*008:01*	*DRB2*002:01*	*DRB3*002:01/* *DRB3*002:02*
HP2.25a	0	11	7	*DQA*gb1:05:03*	*DQA*no4:01*		*DQB*no11:01*	** *DQB*is7:01* **	** *DQB*is1:03* **	*DRA*001:AA*		** *DRB2*004:01* **	
HP2.25b	4	0	7	*DQA*gb1:05:03*	*DQA*no4:01*		*DQB*no11:01*	** *DQB*is7:01* **	** *DQB*is1:02* **	*DRA*001:AA*		** *DRB2*005:01* **	
unHP2.25c	1	0	6	*DQA*gb1:05:03*	*DQA*no4:01*		*DQB*no11:01*	** *DQB*is8:01* **		*DRA*001:AA*		** *DRB2*005:01* **	
pHP2.26	0	2	3				*DQB*no7:01*	*DQB*no4:01*		*DRA*001:01/* *DRA*001:04*	*DRB*no7:01*		
unHP2.27	1	0	10	*DQA*is5:01*	*DQA*gb1:01*	*DQA*is3:01*	*DQB*no11:01*	*DQB*is7:01*	*DQB*is1:02*	*DRA*001:01/* *DRA*001:04*	*DRB1*001:01*	*DRB2*002:01*	*DRB3*002:01/* *DRB3*002:02*
unHP2.28a	1	0	5	*DQA*gb2:02*			*DQB*gb1:02*			** *DRA*001:AA* **	** *DRB*is10:01* **	*DRB2*002:AA*	
unHP2.28b	1	0	4	*DQA*gb2:02*			*DQB*gb1:02*			*DRA*001:01/* *DRA*001:04*	** *DRB*is12:01* **	*DRB2*002:AA*	
unHP2.29	1	0	8	*DQA*no1:01*	*DQA1*002:01:AA*		*DQB*no7:01*	*DQB*no4:01*		*DRA*001:01/* *DRA*001:04*	*DRB1*001:01*	*DRB2*002:AA*	*DRB3*002:01/* *DRB3*002:02*
unHP2.30	1	0	7	*DQA*no2:01*	*DQA*gb1:01*		*DQB*is6:01*	*DQB*is7:01*	*DQB*is1:01*	*DRA*001:AA*		*DRB2*002:AA*	
unHP2.31	1	0	7	*DQA*gb1:05:01*			*DQB*no11:01*	*DQB*is7:01*	*DRB2*002:01*	*DRA*001:AA*		*DRB*no2:01:02*	*DRB3*002:01/* *DRB3*002:02*
unHP2.32	1	0	7	*DQA*gb1:05:02*	*DQA*is2:01*		*DQB*is5:01*	*DQB2*002:AA*		*DRA*001:AA*	*DRB1*001:01*	*DRB*gb2:02*	
unHP2.33a	1	0	8	*DQA*gb2:02*	** *DQA1*002:01:AA* **	** *DQA*gb1:03* **	*DQB*gb1:02*	*DQB2*003:01*		*DRA*001:01/* *DRA*001:04*	*DRB*is12:01*	*DRB2*002:AA*	
unHP2.33b	1	0	8	*DQA*gb2:02*	** *DQA*gb2:01* **	** *DQA*gb1:02* **	*DQB*gb1:02*	*DQB2*003:01*		*DRA*001:01/* *DRA*001:04*	*DRB*is12:01*	*DRB2*002:AA*	
unHP2.34	1	0	7	*DQA*gb2:03*	*DQA2*002:AA*		*DQB1*003:AA*	*DQB*no1:02*		*DRA*001:01/* *DRA*001:04*	*DRB1*002:01*	*DRB2*002:01*	
unHP2.35	1	0	6		*DQA*gb3:01*		*DQB1*003:AA*			*DRA*001:03*	*DRB1*008:01*	*DRB2*002:01*	*DRB3*001:01:02/* *DRB3*001:01:01*
unHP2.36	1	0	6	*DQA*is4:01*			*DQB*is4:01*	*DQB*is9:01*		*DRA*001:01/* *DRA*001:04*	*DRB*is9:01*		*DRB3*002:AA*
unHP2.37	1	0	7	*DQA1*002:01*	*DQA*gb1:02*		*DQB*no9:01*	*DQB2*003:01*		*DRA*001:01/* *DRA*001:04*	*DRB*no7:01*	*DRB2*005:01*	
unHP2.38	0	1	7	*DQA1*002:02*	*DQA2*002:01*		*DQB2*004:01:AA*	*DQB1*004:AA*		*DRA*001:01/* *DRA*001:04*	*DRB*is8:01*	*DRB2*002:01*	*DRB3*002:01/* *DRB3*002:02*
unHP2.39	0	1	8	*DQA*gb1:05:03*	*DQA*no4:01*		*DQB*no7:01*	*DQB*no4:01*		*DRA*001:01/* *DRA*001:04*	*DRB*no7:01*	*DRB2*003:01*	*DRB3*002:01/* *DRB3*002:02*
unpHP2.40	1	0	2							*DRA*001:03*	*DRB*is11:01*		
unpHP2.41	0	1	4				*DQB*no7:01*			*DRA*001:01/* *DRA*001:04*	*DRB*no7:01*	*DRB2*003:01*	

## Data Availability

The raw sequencing data is available on ENA with the accession id PRJEB57925 (Thoroughbred horses) and PRJEB57926 (Icelandic and Norwegian Fjord horses).

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
