# Peer review of "High-Resolution Genotyping of Expressed Equine MHC Reveals a Highly Complex MHC Structure"

_genes, 2023, doi:10.3390/genes14071422_

Round 1
Reviewer 1 Report
Line 24 been not be
Line 134 should the second DRB actually be ..and DQB genes …
Line 237 Eqca-MHC1*g…b…11:02 is the b missing?
you have generated and validated a high resolution and high throughput system for genotyping the MHC in horses, which you indicate is important because it provides novel insights into the repertoire and and structure of the MHC repertoire. can you add some detail in the discussion and summary on how this methodology will have benefits for the health of horse populations?
Reviewer 2 Report
The article is well-written and effectively communicates the objectives, methods, and results of the study. The utilization of high-throughput sequencing technologies for characterizing MHC genes in horses is a significant advancement and holds promise for future research. The comprehensive data generated and the identification of numerous novel sequences contribute to the existing knowledge base.
One area that could be further explored in future studies is the functional implications of the identified novel MHCI and MHCII sequences. Understanding how these newly discovered sequences may influence immune responses and disease susceptibility in horses would provide valuable insights. Additionally, investigating the functional differences and similarities in MHC repertoire among different horse breeds could offer further insights into breed-specific immunological characteristics.
In conclusion, the study successfully employs a MiSeq-based approach to analyze the expressed MHCI and MHCII repertoires in horses. The findings provide important insights into the structure and diversity of the equine MHC repertoire, highlighting the unique features of the MHC in different horse breeds. This research contributes significantly to the field and lays the foundation for future investigations into the functional implications of the identified MHC sequences.
But there is a small problem in this article:
1. While the content of the tables is informative and supports your findings, the visual presentation could be improved to enhance readability and aesthetic appeal.
Reviewer 3 Report
The rise of sequencing as a sophisticated tool to unravel the mysteries of the genome has come along with a massive progress in scientific understanding.
This study takes advantage of a MiSeq to understand the Major Histocompatibility complex (MHC) II gene repertories in selected horse breeds. The experimental approach is based on a well-established design and execution, the data are vast but distributed into subsection to allow a better orientation of the reader.
I do have several comments for consideration:
- The authors state that overall, 168 horses were included. What was the sample size for each breed? What was the age and sex of the animals?
- This leads me to another speculation: are there any possible variations in the MHC system amongst breeds, age or sex?
- The authors could provide a brief scheme depicting the experimental approach in the Material and Methods section.
- The tables hold interesting data sets, however, are very large. I would recommend moving them into Supplementary files while the main document could depict a table or tables summarizing the most important findings.
- Any possible limitations of the study could be discussed in the Discussion section.
- The references in the text should be numbered and placed in square brackets (see Instructions for authors).
- The formatting of the List of references should be revised according to the Journal’s requirements (see Instructions for authors).
Reviewer 4 Report
The paper deals with the equine MHC. It brings a lot of new knowledge, but there are some problems, which must be solved.
Title:
Why twice MHC?
Keywords:
Words from title repeated.
Introduction:
Explain NK (r. 45). BAC (r.79).
Reference for the 1st paragraph; for the rr. 57-72.
Gove some information o the equine genome; MHC – chromosome etc.
MM:
Gove n for Thoroughbreds, Icelandic and Norwegian Fjord Horses.
Throughout the text: check fonts.
Main objection:
The analysis of interest was done, with plenty of new information. However, the presentation is labyrinthine, tables are poorly arranged, as well the comments in the text. The same for Discussion section.
The authors have done excellent work, obtained a huge amount of data. But the data must be sorted logically and clearly. I encourage the team to re-arrange the manuscript in a more concise way.
OK
Round 2
Reviewer 4 Report
The authors followed partly my comments. However, after explanations I agree with the author´s point of view.
Please, check the Table 2A, 2B:
“Number of occurrences in… ?”
“Total number of expressed… ?”
In Table 6, the describing of columns is proper.
I again appreciate the work which has been done.